# Effective and Successful Quantification of Leukemia-Specific Immune Cells in AML Patients’ Blood or Culture, Focusing on Intracellular Cytokine and Degranulation Assays

**DOI:** 10.3390/ijms25136983

**Published:** 2024-06-26

**Authors:** Olga Schutti, Lara Klauer, Tobias Baudrexler, Florian Burkert, Joerg Schmohl, Marcus Hentrich, Peter Bojko, Doris Kraemer, Andreas Rank, Christoph Schmid, Helga Schmetzer

**Affiliations:** 1Department for Hematopoetic Cell Transplantation, Med. III, University Hospital of Munich, 81377 Munich, Germany; olga.schutti@med.uni-muenchen.de (O.S.);; 2Bavarian Cancer Research Center (BZKF), Comprehensive Cancer Center at University Hospital of Augsburg, 86156 Augsburg, Germany; 3Department of Haematology and Oncology, University Hospital of Tuebingen, 72076 Tuebingen, Germany; 4Department of Haematology and Oncology, Red Cross Hospital of Munich, 80634 Munich, Germany; 5Department of Heamatology and Oncology, St.-Josefs-Hospital Hagen, 58097 Hagen, Germany; 6Department of Haematology and Oncology, University Hospital of Augsburg, 86156 Augsburg, Germany

**Keywords:** leukemia-specific assays, immune monitoring, immune modulation, leukemia derived DC, AML

## Abstract

Novel (immune) therapies are needed to stabilize remissions or the disease in AML. Leukemia derived dendritic cells (DCleu) can be generated ex vivo from AML patients’ blasts in whole blood using approved drugs (GM-CSF and PGE-1 (Kit M)). After T cell enriched, mixed lymphocyte culture (MLC) with Kit M pretreated (vs. untreated WB), anti-leukemically directed immune cells of the adaptive and innate immune systems were already shown to be significantly increased. We evaluated (1) the use of leukemia-specific assays [intracellular cytokine production of INFy, TNFa (INCYT), and degranulation detected by CD107a (DEG)] for a detailed quantification of leukemia-specific cells and (2), in addition, the correlation with functional cytotoxicity and patients’ clinical data in Kit M-treated vs. not pretreated settings. We collected whole blood (WB) samples from 26 AML patients at first diagnosis, during persisting disease, or at relapse after allogeneic stem cell transplantation (SCT), and from 18 healthy volunteers. WB samples were treated with or without Kit M to generate DC/DCleu. After MLC with Kit M-treated vs. untreated WB antigen-specific/anti-leukemic effects were assessed through INCYT, DEG, and a cytotoxicity fluorolysis assay. The quantification of cell subtypes was performed via flow cytometry. Our study showed: (1) low frequencies of leukemia-specific cells (subtypes) detectable in AML patients’ blood. (2) Significantly higher frequencies of (mature) DCleu generable without induction of blast proliferation in Kit M-treated vs. untreated samples. (3) Significant increase in frequencies of immunoreactive cells (e.g., non-naive T cells, Tprol) as well as in INCYT/DEG ASSAYS leukemia-specific adaptive—(e.g., B, T(memory)) or innate immune cells (e.g., NK, CIK) after MLC with Kit M-treated vs. untreated WB. The results of the intracellular production of INFy and TNFa were comparable. The cytotoxicity fluorolysis assay revealed significantly enhanced blast lysis in Kit M-treated vs. untreated WB. Significant correlations could be shown between induced leukemia-specific cells from several lines and improved blast lysis. We successfully detected and quantified immunoreactive cells at a single-cell level using the functional assays (DEG, INCYT, and CTX). We could quantify leukemia-specific subtypes in uncultured WB as well as after MLC and evaluate the impact of Kit M pretreated (DC/DCleu-containing) WB on the provision of leukemia-specific immune cells. Kit M pretreatment (vs. no pretreatment) was shown to significantly increase leukemia-specific IFNy and TNFa producing, degranulating cells and to improve blast-cytotoxicity after MLC. In vivo treatment of AML patients with Kit M may lead to anti-leukemic effects and contribute to stabilizing the disease or remissions. INCYT and DEG assays qualify to quantify potentially leukemia-specific cells on a single cell level and to predict the clinical course of patients under treatment.

## 1. Introduction

### 1.1. Current Therapy Strategies for Acute Myeloid Leukemia (AML)

AML is a life-threatening and rapidly progressive malignant disease that goes along with the uncontrolled expansion of myeloid blasts [1]. Treatment of AML is based on high-dose chemotherapy (such as cytarabine and anthracycline), often combined with allogenic stem cell transplantations (SCT) as the only potentially curative therapy [2]. Unfortunately, 80% of successfully (chemotherapy-) and 40% of SCT-treated patients relapse within two years [3,4,5]. For this reason, there is a high need to develop maintenance therapies to stabilize remissions. A promising option to consolidate and control remissions is a targeted therapy with antigen-presenting cells (APCs), such as dendritic cells (DC) [6].

### 1.2. Dendritic Cells for Immune Therapy

As a result of activation and maturation processes, DCs, the most potent antigen-presenting cells, differentiate out of hematopoietic stem cells and regulate complex immune responses via soluble or cellular mediators [6,7]. Myeloid blasts can also be converted into leukemia-derived dendritic cells (DCleu) without inducing the blasts’ proliferation. They generate a strong immune response by presenting the patients’ individual tumor antigens to the immune system and giving rise to leukemia-specific cells that close the gap between the adaptive and innate immune systems [8,9]. These DCleu can be generated ex vivo in cultures and afterwards transferred to patients, or directly in patients’ whole blood (WB), simulating the in vivo situation most since it contains individual patients’ soluble or cellular immune inhibitors or activating components by using combined designated response modifiers such as Kit M: GM-CSF (granulocyte macrophage colony stimulating factor) + PGE1 (prostaglandin E1) [10]. With respect to future immunotherapies, Kit M could convert (residual) blasts to DCleu in vivo and is therefore the focus of research.

### 1.3. Innate and Adaptive Immune Systems

The innate and adaptive immune systems use humoral and cellular mechanisms to defend pathogens and tumor cells [11,12] (Parkin and Cohen, 2001; Dunn et al., 2002). The innate immune system, consisting of macrophages (CD15+), monocytes (CD14+), DCs (CD80+, CD206+, etc.), cytokine induced killer cells (CIK, CD56+CD3+), and healthy natural killer cells (NK, CD56+CD3+), is responsible for the ‘first-line’ response. The adaptive immune system is responsible for a specific and ‘long-term’ immunity, with key players such as B cells (CD19+), T cells (CD3+), and their subpopulations such as naive or non-naive T cells (CD3+CD45RO−: Tnaive, CD3+CD45RO+: Tnon-naive; CD3+CD4+CD45RO+: Tnon-naive; CD4+ and CD3+CD4−CD45RO+: Tnon-naiveCD4), CD4+CD3+ (TCD4+) and CD4-CD3+ (TCD4-) as ‘active’ mediators of immune responses and central memory cells (CD3+CD45RO+CD197+: Tcm) or effector-memory cells (CD3+ CD45RO+ CD197−: Tem) to fight reoccurring antigen.

### 1.4. Intracellular Cytokine Assay (INCYT), Degranulation Assay (DEG), and Cytotoxicity Fluorolysis Assay (CTX)

Targeted antileukemic activity requires a complex interaction of immune cells in the innate and adaptive immune systems. These reactions can be detected and quantified by various leukemia-specific assays (e.g., ELISpot [13,14]., MHC Multimer [15], Cytokine-release [8], or degranulation-based assays). The cytokines IFNy and TNFa play an important role in antitumor immunity [16,17,18] and are known to be induced after contact with antigen-presenting cells [7]. In contrast, blocking INFy causes a significant reduction in immune defense [19,20]. The intracellular cytokine assay (INCYT) allows for the detection and quantification of IFNy- and TNFa-producing cells [21].

In analogy to the INCYT, the DEG allows the detection and quantification, on a single-cell level, of lysosomal-associated membrane glycoproteins (LAMPs) such as CD107a. These glycoproteins are involved in perforin-associated degranulation processes [8], so this assay detects the physiological step after the INCYT assay. Potentially leukemia-specific cell frequencies can be increased after stimulation of immune cells with leukemia-associated antigens (LAA) such as PRAME or WT1, respectively, with staphylococcal enterotoxin B (SEB) for healthy WB samples [22,23,24].

With the CTX, the anti-leukemic effect of (induced) effector cells and viable blasts can be detected and quantified after the influence of stimulator cells (e.g., kit treated vs. untreated WB) [25].

### 1.5. Aims of This Study

The aim of this study was to analyze the qualitative and quantitative suitability of an INCYT and DEG assay to characterize leukemia/antigen-specific immune cells of several lines with or without additional stimulation with leukemia- (in AML patients’) or infectios antigens (in healthy donors’) uncultured blood and, moreover, after stimulation of immune cells after MLC with Kit M-treated (or untreated) WB. Finally, results obtained with the INCYT and DEG were correlated with cells’ antileukemic functionality and patients` clinical data (risk groups and response to induction therapy).

## 2. Results

### 2.1. Prologue

We generated DC/DCleu from leukemic and healthy WB using Kit M-treated and untreated cultures. We analyzed immune cells, focusing on their intracellular cytokine production (IFNy and TNFa) before and after T cell-enriched MLC. Immune cell profiles were correlated with blastolytic functionality, patients’ allocations to ELN-risk groups, and their outcomes. Abbreviations for all subsequently mentioned cell subpopulations are given in Table 1.

### 2.2. Effects of LAA-/SEB-Stimulation in Uncultured (D0) WB (AML and Healthy) on the Intracellular IFNy-Production and Degranulation (CD107a) Positivity of Immune Cell Subtypes

Studying the composition of healthy and leukemic WB samples by INCYT and DEG assays, we found low frequencies of antigen-specific cells of innate and adaptive cell lines in uncultured WB samples. After the influence of specific antigens (LAA (AML) or SEB (healthy)) frequencies of some cell subtypes could be significantly increased (e.g., in AML-samples: INCYT: Tcm, CIK; DEG: Tcm, NK; in healthy samples: DEG: Tnon-naive, Tem, Tcm) (Figure 1).

### 2.3. Effects of Kit M on the Generation of DC/DCleu and Blast Proliferation Using AML and Healthy WB

We observed highly significantly higher frequencies of DC; DCmig; DCleu; and DCleu-mig in the Kit M pretreated vs. untreated control cultures of AML-WB samples (e.g., %DC/cells (Kit M): 25.72 ± 6.98; %DC/cells (Control): 15.13 ± 3.98; *p* ≤ 0.0001). Kit M-treated vs. untreated WB of healthy volunteers also showed significantly higher frequencies of DC and DCmig in the Kit M-treated cultures (Figure 2C). The proliferation of blasts was not induced under Kit M treatment. A highly significant decrease in frequencies of proliferating blasts was found in Kit M-treated cultures (e.g.; %Blaprol-CD71/Bla (Kit M): 6.98 ± 4.81; %Blaprol-CD71/Bla (Control): 10.19 ± 5.47; *p* < 0.0001) (Figure 2A,B).

### 2.4. Effects of Kit M-Treated vs. Untreated AML WB before and after T Cell-Enriched MLC on Immunoreactive Cells

We observed generally higher frequencies of immune cells after mixed lymphocyte culture (MLC) after culture (control) compared to uncultured T cells enriched in MLC, which can be attributed to the effect of IL-2 (Figure 3A) (Rosenberg, 2014). We found highly significantly higher frequencies of Tprol-early, Tprol-late, and Tnon-naive in Kit M-treated vs. untreated MLC fractions (e.g., %Tnon-naive/T (Kit M): 69.24 ± 15.84; %Tnon-naive/T (control): 63.65 ± 17.01; *p* = 0.0004) (Figure 3B). Other cell subsets such as T4-, Tcm, and Tem showed no significant differences between the groups compared. MLCs of healthy WB samples showed comparable results.

### 2.5. Composition of IFNy- and TNFa-Producing Immune-Reactive Cells after T Cell-Enriched MLC, Using Leukemic WB with and without (Control) Kit-Pretreatment (Kit M) and with (+LAA) and without (-LAA) LAA-Stimulation

In Kit M pretreated vs. untreated cells after MLC (without LAA stimulation), we found highly significantly higher frequencies of INFy-producing cells of several lines in Kit M pretreated cultures (e.g., %CIK_IFNy_/CIK (Kit M): 48.37 ± 22.7; %CIK_IFNy_/CIK (Control): 16.91 ± 9.75; *p* = 0.00005) (Figure 4A). Frequencies of TNFa-producing immune cells after Kit M-treated vs. untreated MLC showed comparable results (e.g., %T_TNFa_/T (Kit M): 23.74 ± 15.91; %T_TNFa_/T (Control): 11.94 ± 8.03; *p* = 0.0002; %CIK_TNFa_/CIK (Kit M): 42.36 ± 23.66; %CIK_TNFa_/CIK (Control): 17.90 ± 13.64; *p* = 0.0002) (Figure 4C). The frequencies of antigen-specific cells, as detected by the INCYT assay, were comparable with or without LAA stimulation in the Kit M pretreated vs. not pretreated MLC cell fraction (Figure 4A–D). Moreover, frequencies of antigen-specific cells (with or without SEB stimulation) in Kit M pretreated vs. not pretreated cell fractions were comparable to cell compositions from AML patients.

### 2.6. Effects of Kit M-Treated vs. Untreated AML WB on the Degranulation of Immune Cells after MLC

Without LAA stimulation, we observed highly significantly higher frequencies of degranulating CD107a-positive cells in Kit M-treated vs. untreated cells after MLC (e.g., %NK_107a_/NK (Kit M): 30.91 ± 19.72; %NK_107a_/NK (Control): 18.21 ± 14.21; *p* = 0.0012; %B_107a_/B (Kit M): 33.74 ± 20.42; %B_107a_/B (Control): 28.02 ± 15.27; *p* = 0.012) (Figure 5A). The results of LAA-stimulated vs. unstimulated cell fractions were comparable (Figure 5A,B). We found antigen-specific cells in healthy WB in Kit M pretreated vs. not pretreated (with or without SEB stimulation) cell fractions.

### 2.7. Effects of Kit M-Treated WB on the Antileukemic and Blastolytic Activity after MLC, Detected via CTX

Applying a CTX fluorolysis assay blast lysis was observed in 82% of Kit M pretreated cases after 3 h and in 100% of all cases after 24 h of coincubation of effector cells with target cells. In the control group, blast lysis was observed in 65% of cases after 3 h and in 71% after 24 h of coincubation (Figure 6A). Comparing cases with vs. without Kit M treatment, improved lysis vs. control was demonstrated in 100% of cases, both after 3 h and after 24 h of coincubation of the target with effector cells (Figure 6B). A highly significant increase in blast lysis was seen in the Kit M-treated vs. untreated group after 3 h of coincubation with target cells (29.96 vs. 5.66% of lysed blasts, *p* = 0.001). Selecting the best achieved lysis in the Kit M pretreated vs. not pretreated group, a blast lysis of 47.58% vs. 12.78% (*p* = 0.00002) was seen (Figure 6D).

### 2.8. Correlation of the Relative Improvement of IFNy and TNFa Secretion and the Degranulation with the Relative Improvement of Blast Lysis in Kit M-Treated (vs. Untreated) WB after MLC

We demonstrated highly significant correlations between induced INFy or TNFa-producing immune cells (in Kit M pretreated vs. untreated cell fractions) and improved blast lysis (e.g., Tem_INFy_, T_TNFa_) (Figure 7A,B), but not with frequencies of NK_IFNy_, Tcm_IFNy_, and T4-_TNFa_, NK_TNFa_. Moreover, we detected a significant positive correlation between degranulating T4+_107a_ (r = 0.69), Tcm_107a_ (r = 0.83), and a borderline significant positive correlation between degranulating T_107a_ (r = 0.76), and B_107a_ (r = 0.83), with improved blast lysis (Figure 7C). No correlations were found for T4-_107a_ and NK_107a_ values and improved blast lysis.

### 2.9. Cases with Intracellular IFNy Production before and after MLC, Subdivided into Cytogenetic ELN Risk Groups (Favorable and Adverse)

Cases at first diagnosis allocated to favorable vs. adverse cytogenetic risk according to ELN risk allocation were characterized by higher frequencies of INFy-producing cells (e.g., %Tcm_INFy_(favorable risk): 9.44 ± 3.53; %Tcm_INFy_(adverse risk): 2.90 ± 6.76) although differences were not significant due to low case numbers (Figure 8A). After MLC, no significant differences were found in Kit M pretreated compared to the untreated results of the control group and grouped according to the ELN risk groups (favorable and adverse risk) (Figure 8B). Due to low case numbers, DEG analyses of patients at first diagnosis were not possible.

### 2.10. Cases with Intracellular IFNy Production before and after MLC, Subdivided into Patients’ Response to Induction Chemotherapy

Patients at first diagnosis who had responded vs. not responded to induction chemotherapy were characterized by higher frequencies of INFy-producing cells, although differences were not significant due to low case numbers (Figure 8C). After the MLC and adjusting the Kit M data with the control values, we compared the values of the two groups (responders and non-responders): we saw no differences in IFNy production in the group of responders vs. non responders to induction chemotherapy (Figure 8D). Due to low case numbers, DEG analyses of patients with or without a response to induction chemotherapy were not possible.

## 3. Discussion

### 3.1. Use of DC-Based Immunotherapy in AML

Dendritic cells (DCs) capturing, processing, and presenting antigens in a co-stimulation manner to immune cells play a crucial role in initiating and regulating immune responses. DC-based immunotherapy for AML uses DCs loaded with tumor antigens, or DC (DCleu), which were shown to migrate to tissues and enhance leukemia-specific immune responses in the body [28,29]. DCleu present individual patients’ leukemic antigen repertoire to immune cells, leading to robust and targeted immune responses, without the need for the loading of leukemic antigens on (monocyte derived) DCs [26,30,31,32].

### 3.2. Generation of DCs Ex Vivo

Studies have shown that DCleu-based immunotherapies induce potent anti-leukemic immunity both in vivo and ex vivo. Here we demonstrated (as shown before) that Kit M (GM-CSF and PGE-1) gives rise to DC/DCleu subtypes without induction of blasts’ proliferation ex vivo [8,26,29] (A,B). As previously demonstrated, we observed increased frequencies of immune cells after T cell-enriched MLC vs. before MLC (A). These findings are due to the effects of IL-2 stimulation, as shown before [33,34]. In WB, treatment with Kit M resulted in significantly higher frequencies of T cell subsets (e.g., Tprol-early, Tnon-naive) as well as innate (NK, CIK) cells after MLC compared to the control group (Figure 3B). This confirms previous findings [8,9].

### 3.3. Different Age Groups in AML Patients’ WB Samples and Healthy Donors

The average age difference between the samples of AML patients and the healthy controls in our cohort is 33.2 years. Other groups have shown that the age of individuals has an impact on the blood composition (hematocrit, hemoglobin, etc.) and cells’ morphology [35]. However, it was also shown that the frequencies of T cells (CD4+ and CD8+), as well as of B- and NK cells, remained relatively stable throughout adulthood [36]. We found that the immunocytological composition of AML and healthy samples was comparable (without showing age-specific differences); nonetheless, due to the ambivalent state of research, we decided against a direct comparison of the WB samples from (young) healthy volunteers and (elderly) AML patients.

### 3.4. Detection of Leukemia-Specific Cells

The **ELISpot Assay** (Enzyme Linked Immuno Spot Assay) quantifies cytokine secreting cells and has become the standard assay to detect leukemia-specific cells since its introduction in 1983 by Sedgwick and Holt [13,14]. Additional stimulation, e.g., with INFy, can be carried out to increase the HLA expression of target cells. Cytokine producing immune cells are detected with antibodies as ‘spots’, that can be counted (although not assigned to defined immune cell subtypes) and quantified compared to controls [37,38].

The **MHC multimer technology** allows a reliable prediction of MHC-binding peptides and potential T cell epitopes that can be monitored [15]. Primarily, CD8+ T cells are high-impact mediators of antigen-specific immunity. To identify relevant immunological correlates, additional phenotypic characteristics of immune cells need to be considered [39]. Problematic in this procedure are the interactions between the pMHC and the T cell receptor, especially when using the pMHC multimers to stain anti-cancer-directed T cells tendentially bearing lower affinity T cell receptors [40]. Since CD4+ antigen-specific cells, which also play an important role in the development of the immune response, cannot be quantified by this technology, we have decided against the application of this procedure for our study [41].

**Intracellular cytokine staining** (INCYT) detects both the phenotype of individual cells and the cytokines (INFy, TNFa) they produce intracellularly and is used to quantify immune cells’ reactivity, particularly in vaccination trials to assess immunogenicity (Lovelace et al., 2011). The retention of cytokines within cells not only accelerates their detection compared to other methods but also enables the identification of the responsible cells (e.g., Tcm, Tem, Treg, NK, and CIK cells [16]. While most routine work uses 4–5 different colors (fluorochrome-stained specific antibodies), optimized protocols have been developed for analyzing up to 18 colors. The INCYT can be applied to quantify (gated) specific cell subsets in various cell suspensions by flow cytometry [42]. Additionally, INCYT is a fast and cost-effective technique and a method with no need for radioactive compounds such as traditional assays such as 3H-thymidine incorporation used in proliferation assays or ^51^Cr release cytotoxicity assays [41,43,44].

Among these methods, INCYT provides the greatest potential for analyzing (simultaneous or single) cytokine production (INFy or TNFa) in various immune-reactive cells, as it allows for immunophenotyping and simultaneous detection of cytokine production in different cell subtypes such as Tcm, Tem, Treg, NK, and CIK cells [8,16,45].

Similar to the INCYT, the **Cytokine Secretion Assay (CSA)** quantifies IFNy-producing and releasing cells with high sensitivity [8,46]. Although it may provide a higher sensitivity compared to the INCYT (particularly when dealing with low frequencies of IFNy-positive cells) [44,46], we and others have found a significant correlation between the IFNy production obtained by the CSA and the INCYT assays [8,47]. The disadvantages of the CSA are the high cell need and the fact that it can quantify only INFy-producing cells, not allowing simultaneous analysis of several cytokines [48].

The potential cytotoxic capacity of effector cells can be assessed using a **degranulation assay (DEG)** that detects the transport of lysosome-associated membrane protein-1 (LAMP-1, CD107a) to the cell surface on a single cell level in several lines of immune cells [49]. CD107a is used as a functional marker for the identification of (degranulating), e.g., natural killer cell activity [50]. The advantages and pitfalls of the DEG (compared to the methods already mentioned) are comparable to those of the INCYT. The DEG quantifies the degranulation activity of defined cells shortly before their mediation of cytotoxicity (which can be quantified by a cytotoxicity assay). To determine if DC/DCleu-stimulated immunoreactive cells were further activated by the addition of **LAA (AML) and SEB (healthy)**, we added LAA WT-1 and PRAME to uncultured WB and WB after MLC with and without Kit M treatment of AML samples and, in analogy, SEB to healthy samples. The addition of LAA to uncultured WB of AML-patients significantly increased the INFy production of CIK cells and the degranulation activities, e.g., of Tem. The addition of SEB in the WB of healthy volunteers significantly increased the degranulation activity of T cells (e.g., T4−, T4+, Tnon-naive, Tem, and Tcm) (Figure 1). The addition of LAA/SEB after MLC did not significantly increase the frequencies of IFNy-secreting or degranulating cells either in AML WB-samples nor in healthy WB-samples (Figure 4 and Figure 5). In our previous work, we had already shown that results obtained with Kit-pretreated (DC/DCleu-containing cells) as stimulator cells in the MLC were comparable with those obtained after LAA stimulation: no increased frequencies of leukemia-specific cells were seen [8], pointing (again) to the stimulatory (DC/DCleu-inducing) effect of Kit M, leading to increased production of leukemia-specific cells after MLC (without additional LAA stimulation).

Critical to flow cytometric quantification of (specific) cells is that the expression of antigens on the cell surface or in intracellular organelles can vary in intensity. This may be due to low antigen expressions or, among other things, to the fact that the development and provision of (antigen-specific) cells under stimulation in culture or during the course of a patient’s disease is a continuous process, increasing cell frequencies with a certain phenotype under certain influences. Therefore, the quantification of specific cell subtypes must always be carried out in the context of controls (e.g., cell proportions before ex vivo/in vivo treatment, isotype controls) [51].

To determine whether a DC/DCleu-mediated blast kill follows increased intracellular cytokine production (INCYT) and increased degranulation (DEG) after MLC, we used a non-radioactive **cytotoxicity fluorolysis assay** (CTX). By co-culturing target cells (e.g., blasts) with effector cells (previously stimulated or unstimulated with DC/DCleu), it allows the effector cells to induce apoptosis and subsequently lyse the target cells. After an incubation time of 0, 3, and 24 h, viable blasts labeled with fluorochrome-labeled antibodies can be quantified. Cases with/without lysis or with/without improved blast lysis compared to control can be evaluated, and moreover, the efficiency of achieved lysis (e.g., %lysed/increased blasts) be evaluated [33].

### 3.5. Leukemia-Specific Activity of Immune-Reactive Cells

**Leukemia-specific activity (or cytotoxicity)** involves a complex synergy of the innate and adaptive cellular immune systems, as well as the humoral immune system. Various innate and adaptive immune cells exert cytotoxicity through different mechanisms: early and rapid release of cytolytic molecules such as perforin and granzyme by degranulation, or later and slower interaction with the Fas ligand (FasL) and TNF-related apoptosis-inducing ligand (TRAIL), and/or secretion of tumor necrosis factor alpha (TNFa) or interferon gamma (IFNy). IFNy plays a diverse role: it is crucial for regular immunity, promoting both innate and adaptive immune responses, licensing immune cells to exert cytotoxicity, which is strongly associated with anti-tumor immunity, and facilitating tumor surveillance, recognition, and elimination.

During the early phase of an immune response, IFNy is predominantly secreted by innate immune cells such as NK cells, iNKT cells, DCs, and macrophages upon activation. It not only enhances the effector mechanisms and IFNy secretion of innate immune cells but also aids in tumor recognition and elimination. By upregulating the major histocompatibility complex (MHC) class I and II antigen- and cross-presentation pathways, as well as MHC-II expression on classically non-MHC-II-expressing cells, it enhances tumor cell susceptibility to adaptive immune cells. Additionally, by upregulating the expression of various apoptotic pathway molecules and downregulating anti-apoptotic molecules, it increases tumor cell susceptibility to extrinsic and intrinsic pathways of apoptosis. Importantly, by promoting the differentiation, proliferation, and activation of type 1 helper T cells (Th1) and cytotoxic T cells (Tc), IFNy bridges innate and adaptive immunity, initiating the advanced phase of an immune response. During the advanced phase of an immune response, IFNy is primarily secreted by adaptive immune cells such as Th1 and Tc cells upon primary activation by DCs or secondary activation by their specific antigen [8,52,53].

Under certain circumstances, IFNy can also stimulate immune-suppressive mechanisms in tumor cells, leading to immune escape, although it inhibits the differentiation of type 2 helper T cells (Th2) and of regulatory T cells.

In our previous work, we showed higher frequencies of INFy-producing regulatory T cells (Treg) in AML vs. healthy uncultured cells,; however, there was a downregulation of INFy Tregs in Kit M pretreated (vs. not pretreated) AML patients’ WB [34].

In this study, we investigated frequencies of leukemia-specific cells in uncultured AML-patients’ immune cells (with and without additional stimulation with LAA) and, moreover, the effects of Kit M pretreated whole blood (WB) after MLC on the generation of leukemia-specific cells, with a focus on DEG and INCYT (INFy and TNFa) assays, followed by a correlation of results with cytotoxicity assays and diagnostic/clinical data. Using the CTX, we could confirm the data already shown before: we could demonstrate a significantly improved blast lysis activity after Kit M (vs. without Kit M) pretreated WB MLC (Figure 6).

The use and application of these leukemia/antigenspecific cells will contribute in the future to detecting and quantifying the (leukemia-/antigen-specific) activation and provision of several immune cells of different lines to fight infections or cancer, e.g in the context of several immunotherapies. Especially the role of several NK cell subtypes (positive for CD56, CD16, CD57, etc.) in the mediation of immune reactions might contribute to a better understanding of reaction profiles in immunotherapies [54,55].

### 3.6. Correlation of Results of CTX Values and Leukemia-Specific Assays such as INCYT (INFy and TNFa) and DEG

With respect to a potential clinical application of Kit M, it is crucial to evaluate whether Kit M pretreatment of WB leads to an improved mediation of (leukemia-specific, antileukemic) functionality mediated by specific cell subtypes. We show a correlation of IFNy- and TNFa-secreting adaptive and innate immune cells (e.g., T_INFy_, T4+_INFy_, CIK_INFy_, and Tem_INFy_) and degranulating immune cells (e.g., T_107a_, T4+_107a_, Tcm_107a_, B_107a_) with improved blast lysis (Figure 7). In part, these data confirm previous data obtained with CSA-assays [8] and point to the fact that induced leukemia-specific cells are responsible for a Kit-mediated blast kill [56,57]. These data also contribute to a refined assessment of the role of various cellular subtypes in the mediation (or suppression) of antileukemic reactions [8,34], especially with respect to evaluating the immunological influences of new therapies. Here we can clearly confirm that Kit M treatment of leukemic WB (resembling in vivo stimulations most) increases (leukemia-specific) activated immune cells and leads to improved blast lysis compared to control. This could point to potential in vivo efficacy against blasts after Kit M treatment.

Our data also show that functional assays (INCYT, DEG) allow prognostically relevant evaluations: uncultured cells from patients with favorable vs. adverse ELN risk types, as well as patients with vs. without response to induction therapy, were characterized by higher frequencies of leukemia-specific cells. These results point to the functional role of these cells in suppressing or deleting leukemic cells [58]. Kit M treatment (vs. without treatment) does not support these findings, which could mean that Kit M treatment of patients (in vivo) leads to improved antileukemic reactivity independent of patients’ ELN risk or response to induction therapy).

## 4. Material and Methods

### 4.1. Sample Acquisition

Sample collection of whole blood (WB) was conducted after obtaining the written informed consent of blood donors in accordance with the Declaration of Helsinki and the ethical committee of the LMU in Munich (No. 33905). The heparinized peripheral WB-samples and clinical reports were provided by the University Hospitals of Augsburg, Oldenburg, and Munich, as well as the Diakonieklinikum Stuttgart, the Rotkreuzklinikum Munich, and the St. Josefs Hospital Hagen.

### 4.2. Patients’ Characteristics

Samples were collected from 26 AML patients and 18 healthy volunteers. On the day of sample collection, patients were on average 62.5 years old (range 29–98) years old, healthy individuals were 29.3 years old (range 17–58) years old. For patients, the female-to-male ratio was 1.4:1 and 1:1.7 for the healthy ones. The patient’s peripheral WB contained on average 37.8 (range 7–79) % blasts, as detected by flow cytometric analyses. The patients were classified by the French American British (FAB) classification (Bennett et al., 1976), the etiology (primary and secondary AML), the stage of disease (first diagnosis, relapse), the blast phenotype, blast frequencies in peripheral blood, and the cyto- and molecular genetics by the European Leukemia Net (ELN) classification [3]. An overview is given in Table 2.

### 4.3. Sample Preparation

The cellular composition of WB samples from AML patients and healthy volunteers was evaluated in uncultured WB. The DEG and the INCYT were performed on uncultured WB samples to quantify potentially antigen- or leukemia-specific cells in uncultured WB. Dendritic cells were cultured from WB, and mononuclear cells (MNC) were isolated. Therefore, Ficoll-Hypaque density gradient centrifugation and a separating solution with a density of 1.077 g/mL (Biocoll-Separating-solution, Biochrom, Berlin, Germany) were used. T cells were isolated from MNC via MACS MicroBead Technologie (Miltenyi Biotec, Bergisch Gladbach, Germany) and frozen for later usage. For the freezing process, a medium with 70% RPMI-1640 medium (Biochrom, Berlin, Germany), 20% human serum (HealthCare Europa GmbH, Vienna, Austria), and 10% dimethyl sulfoxide (Sigma Aldrich Chemie GmbH, Steinheim, Germany) was used and stored at −80 °C until utilization [59].

### 4.4. Characterization and Quantification of Cells via Flow Cytometry

Flow cytometric analyses were performed to assess and quantify the frequencies and phenotypes of cell subpopulations of leukemic blasts, B cells, DCs, monocytes, T cells, NK cells, iNKT cells, and CIK cells and their functionality. These analyses were performed before and after cell culture. Abbreviations for all cell types are given in Table 1.

For cell staining, various monoclonal antibodies (moAbs) labeled with Fluorescein Isothiocyanate (FITC), Phycoerythrin (PE), Phycoerythrin Cyanine7 (PC7), or Allophycocyanin (APC) were used. Antibodies were provided by Beckman Coulter (Krefeld, Germany), Becton Dickinson (Heidelberg, Germany), Miltenyi Biotec (Bergisch Gladbach, Germany), BioLegend (Koblenz, Germany), and Santa Cruz Biotechnology (Heidelberg, Germany). For analyses the following were used: FITC-labelled moAbs CD3, CD4, CD15, CD45RO, CD65, CD71, CD161, IPO38; PE-labelled moAbs CD3, CD4, CD34, CD56, CD65, CD80, IFNy; PC7-labelled moAbs CD3, CD4, CD14, CD33, CD34, CD65, CD117, CD197, TNFa; and APC-labelled moAbs CD3, CD14, CD19, CD34, CD56, CD69, CD117, CD206. 7AAD was used to detect non-viable cells. To detect cell degranulation, a FITC-conjugated moAb CD107a, was used. Additionally, for intracellular cytokine (e.g., IFNy, TNFa) staining or to detect IPO38 pos. cells fixation and permeabilization were conducted using Medium A and Medium B (FiX&PERM, Thermo Fisher Scientific, Waltham, MA, USA). In accordance with the manufacturer`s instructions, isotype controls were included. All flow cytometric measurements were conducted using the fluorescence-activated cell sorting flow cytometer FACSCalibur (Becton Dickinson) and the analysis software CellQuestPro (Becton Dickinson, version 5.1), using a uniformly defined gating strategy [8].

### 4.5. Dendritic Cell Culture (DCC)

Dendritic cells (DC) and leukemia-derived dendritic cells (DCleu) were generated by treating WB with “Kit M”, containing 800 U/mL Granulocyte-Macrophage-Colony-Stimulating Factor (GM-CSF; Sanofi-Aventis, Frankfurt, Germany) and 1 μg/mL Prostaglandin-E1 (PGE1; Santa Cruz Biotechnology).

Therefore, WB was cultivated in sterile 24-well plates (CellstarÒ, Greiner bio-one, Kremsmuenster, Austria) using 500 μL X-VivoTM 15 medium (Lonza, Verviers, Belgium) and 500 μL WB. “Kit M” was added on day 0 and after 2–3 days, as described (Schwepcke et al., 2022). Untreated WB samples served as controls. DC cultures were incubated at 37 °C, 21% O_2_, and 10% CO_2_ for 7 days.

Flow cytometric analyses of DC-subtypes and proliferating blasts from both Kit M-treated WB (WBDC(M)) and untreated WB (WBDC(Control)) were performed before and after culture using a refined gating strategy [8]. In cases of less than 1% blasts in the cell fractions, DCleu and associated subgroups could not be evaluated.

### 4.6. Mixed Lymphocyte Culture (MLC)

DC/DCleu cultures were used as stimulators to activate T cell-enriched immune cells after MLC. Therefore, 2.5 × 10^5^ cells from WBDC (Control)/(Kit M) were co-cultured with 1 × 10^6^ previously thawed autologous T cells, diluted in 1 mL RMPI-1640 medium containing 100 U/mL penicillin (Biochrom) and 15% human serum, and incubated at 37 °C, 11.21% O_2_ and 10% CO_2_. 50 U/mL Interleukin-2 (IL-2, PeproTech, Berlin) was added to all cultures on day 0 and again after 2–3 and 5–6 days. Cells, later referred to as WBDC-MLC (Control)/(M), were harvested after 7 days.

Flow cytometric analyses of different immune cell subtypes were performed before and after MLC [10].

### 4.7. Degranulation Assay (DEG) and Intracellular Cytokine Assay (INCYT)

As a marker for induced cell cytotoxicity, cell degranulation was quantified in uncultured WB in DCC-MLC (Control)/(M) using a FITC-conjugated antibody against CD107a. Uncultured WB was stimulated in parallel with leukemia-associated antigens (LAA): 2 μg/mL “Wilms Tumor 1” (PepTivator^®^WT1, Miltenyi Biotec) and 2μg/mL “Preferentially Expressed Antigen of Melanoma” (PepTivator^®^PRAME, Miltenyi Biotec); unstimulated WB served as a control. Healthy WB samples before and after MLC were stimulated with 10 µg/mL staphylococcal enterotoxin B (SEB, Sigma-Aldrich, St. Louis, MO, USA); unstimulated healthy WB served as a control. To avoid loss or weakening of CD107a antibodies´ fluorescence, 2 μg/mL Monensin solution (BioLegend) was added to the cultures. After an incubation of 16 h at 37 °C, 21% O_2_ and 10% CO_2_ cells were harvested, stained, and analyzed by flow cytometry [34,60].

### 4.8. Intracellular Cytokine Assay (INTCYT)

To quantify the intracellular production of Interferon-y (IFNy) and tumor necrosis factor-a (TNFa) by different immune cells in uncultured WB and in WB after MLC, with and without Kit M stimulation, the intracellular cytokine assay was performed. In analogy to the DEG, only uncultured WB were stimulated with LAA or SEB. To avoid spontaneous cytokine secretion, a 5 μg/mL Brefeldin A solution (BioLegend) was added. All cultures were incubated for 16 h at 37 °C, 21% O_2_, and 10% CO_2_. After harvest, intracellularly produced IFNy and TNFa were evaluated (as described in Chapter 2.3; [8]).

### 4.9. Cytotoxicity Fluorolysis Assay (CTX)

The cytotoxicity fluorolysis assay was conducted to assess the lytic activity of T cell-enriched immunoreactive cells in WBDC-MLC (Control)/(M) (“effector cells”) against leukemic blasts (“target cells”). Therefore, effector and target cells (with a ratio of 1:1) were co-cultured in a medium containing 85% RPMI/PS and 15% human serum and incubated for 3 and 24 h at 37 °C, 21% O_2_, and 10% CO_2_. Target cells were stained with the respective antibodies before incubation. After harvest, 7AAD and a defined number of fluorosphere beads (Beckman Coulter) were added. As a control, effector and target cells were cultured separately and mixed shortly before measurements.

Flow cytometric analyses were performed after 3 and 24 h using a refined gating strategy [33]. The lytic activity against blasts (“blast lysis”) is defined as the difference in frequencies of viable blasts in the effector-target cell cultures as compared to controls, and “improved blast lysis” is defined as the difference in proportions of “blast lysis” achieved in WBDC-MLC (M) as compared to WBDC-MLC (Control).

### 4.10. Statistical Methods

All cell measurements were performed using flow cytometry (FASCCalibur, Becton Dickinson, Heidelberg, Germany) and the software BD CellQuestPro (Becton Dickinson, version 5.1, Heidelberg, Germany), as described in Chapter 2.4. The data obtained from this was further processed with Excel (Microsoft, version 2404, Redmond, WA, USA) and PRISM 9 (GraphPad Software, Software version number 9.5.1, San Diego, CA, USA). Statistical Methods: the data is presented as mean ± standard deviation. Statistical comparisons of two groups were performed using the one-tailed *t*-test and the Pearson correlation coefficient. The strength of the relationship between two variables was divided into: *p* values > 0.1 = ‘not significant’; *p* values between 0.1 and 0.05 = ‘borderline significant’; *p* values ≤ 0.05 = ‘significant’; *p* values ≤ 0.005 = ‘highly significant’.

## 5. Conclusions

We could show that functional assays (DEG, INCYT, and CTX) qualify to detect and quantify leukemia-specific/antileukemic immunoreactive cells at a single-cell level (in uncultured and cultured settings) and assess the impact of Kit M on the provision of leukemia-specific/leukemia-cytotoxic immune cells after T cell-enriched MLC.

These single cell-based assays are superior to demonstrate specifically induced reactions against blasts compared to, e.g., HLA blocking experiments.

We recommend LAA/SEB stimulation of uncultured WB cells to increase potentially antigen-specific cell fractions (what is not necessary in Kit M pretreated DC/DCleu-containing WB samples after MLC).

We also conclude that Kit M in vivo treatment of leukemic patients might induce (DC/DCleu mediated) antileukemic reactions.

## Figures and Tables

**Figure 1 ijms-25-06983-f001:**
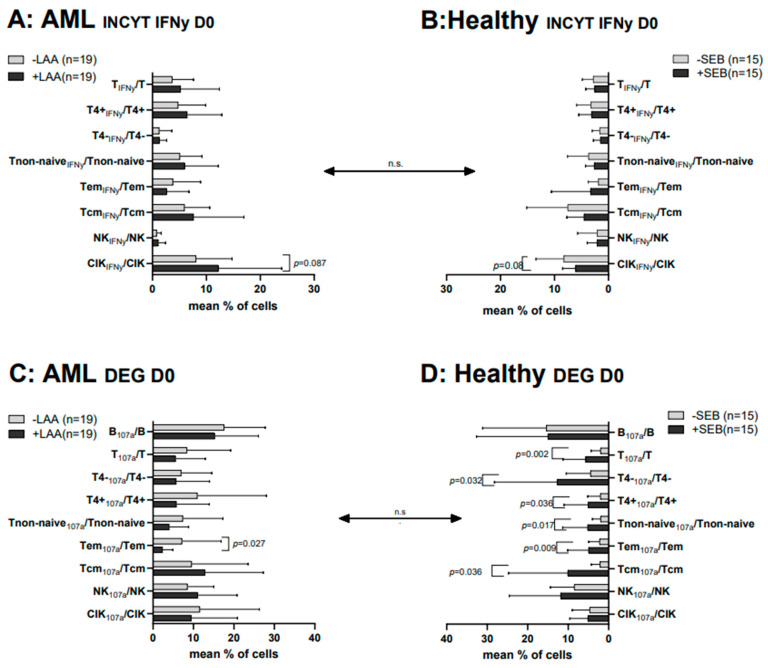
Effects of LAA/SEB-stimulation in uncultured WB (AML and healthy) on the intracellular IFNy-production and degranulation of various T/B and innate immune cell subsets. The stimulatory effects of LAA stimulation on the provision of intracellularly IFNy production (leukemia-specific) immunoreactive cells and subsets in untreated WB of AML patients (**A**) and, in addition, the effects of SEB-stimulation on untreated WB of healthy volunteers on the provision of antigen-specific intracellularly INFy production (**B**) are given. The stimulatory effects of LAA stimulation on the provision of degranulating (leukemia-specific) immunoreactive cells and subsets in untreated WB of AML patients (**C**) and, in addition, the effects of SEB-stimulation on untreated WB of healthy volunteers on the provision of the degranulation (**D**) are given. Given are the mean frequencies ± standard deviation (SD) of T/B cells and innate immune cells. Statistical significance was tested by a multiple-paired *t*-test. (n) number of cases; significance is defined as “highly significant” in cases with *p*-values ≤ 0.005, “significant” with *p*-values ≤ 0.05, “borderline significant” with *p*-values between 0.05 and 0.1, and “not significant” (n.s.) with *p*-values ≥ 0.1. Abbreviations of cell subpopulations are given in Table 1.

**Figure 2 ijms-25-06983-f002:**
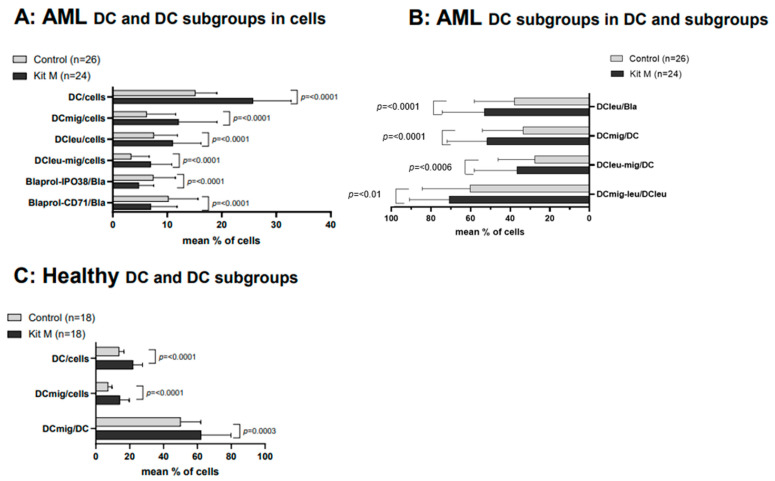
Effects of Kit M on the generation of DC/DCleu and blast proliferation using AML and healthy WB. WB samples of AML patients (**A**,**B**) and healthy volunteers (**C**) were cultured with and without Kit M treatment as controls for 7 days. Given are the mean frequencies ± standard deviation (SD) of DC-subtypes in leukemic WB cells (**A**), subgroups (**B**), and healthy WB cells and DCs (**C**). Statistical significance was tested by a multiple-paired *t*-test. (n) number of cases; significance is defined as “highly significant” in cases with *p*-values ≤ 0.005, “significant” with *p*-values ≤ 0.05, “borderline significant” with *p*-values between 0.05 and 0.1. Abbreviations of cell subpopulations are given in Table 1.

**Figure 3 ijms-25-06983-f003:**
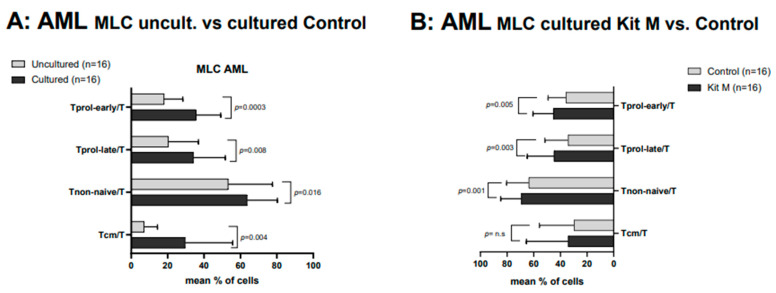
Effects of Kit M-treated and untreated AML WB before and after autologous T cell-enriched MLC on the composition of immunoreactive cells. Immune cells in AML patients’ WB were cultured with vs. without Kit M as a control for 7 days and quantified before (**A**) and after (**B**) MLC. Cells in uncultured vs. cultured control cells (left side) as well as cells after culture with vs. without previous Kit M treatment (right side) are given. Given are the mean frequencies ± standard deviation (SD) of T cells and subtypes. Statistical significance was tested by a multiple-paired *t*-test. (n) number of cases; significance is defined as “highly significant” in cases with *p*-values ≤ 0.005, “significant” with *p*-values ≤ 0.05, “borderline significant” with *p*-values between 0.05 and 0.1, and “not significant” (n.s.) with *p*-values ≥ 0.1. Abbreviations of cell subpopulations are given in Table 1.

**Figure 4 ijms-25-06983-f004:**
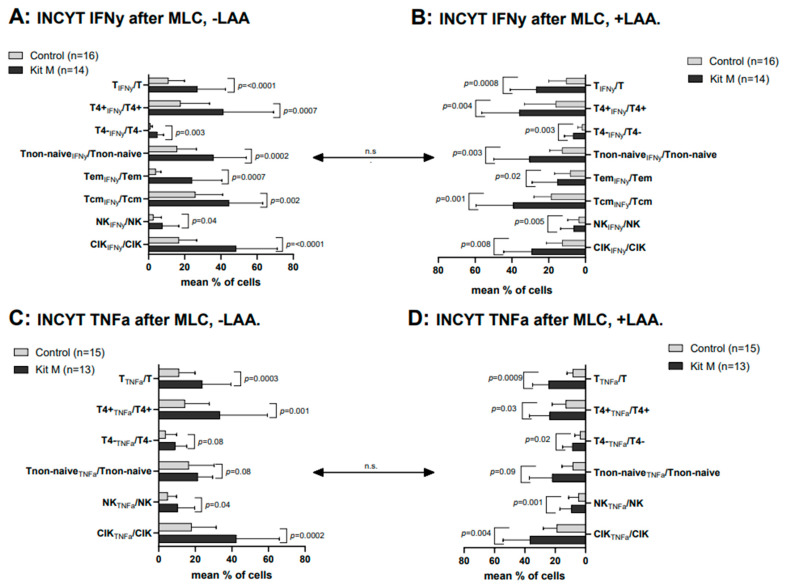
Composition of IFNy- and TNFa-expressing immune-reactive cells after T cell-enriched MLC, using Kit M pretreated vs. not pretreated leukemic WB with (+LAA) and without (−LAA) LAA-stimulation. The effects of Kit M treatment vs. without treatment as control on the provision of intracellularly IFNy (**A**,**B**) and TNFa (**C**,**D**) production (leukemia-specific) immunoreactive cells and subsets in the WB of AML patients after MLC are given. In addition, the stimulatory effects of LAA stimulation (**B**,**D**) vs. without LAA stimulation (**A**,**C**) on the provision of intracellularly IFNy (**A**,**B**) and TNFa (**C**,**D**) production (leukemia-specific) immunoreactive cells and subsets in the WB of AML patients after MLC are given. Given are the mean frequencies ± standard deviation (SD) of T cells and innate immune cells. Statistical significance was tested by a multiple-paired *t*-test. (n) number of cases; significance is defined as “highly significant” in cases with *p*-values ≤ 0.005, “significant” with *p*-values ≤ 0.05, “borderline significant” with *p*-values between 0.05 and 0.1, and “not significant” (n.s.) with *p*-values ≥ 0.1. Abbreviations of cell subpopulations are given in Table 1.

**Figure 5 ijms-25-06983-f005:**
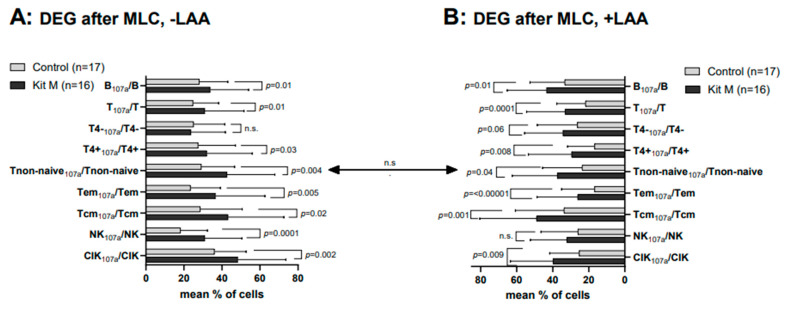
Composition of degranulating immune-reactive cells after T cell-enriched MLC, using Kit M pretreated vs. not pretreated leukemic WB with (+LAA) and without (−LAA) LAA stimulation. The effects of Kit M treatment vs. without treatment as control on the provision of degranulating (leukemia-specific) immunoreactive cells and subsets in the WB of AML patients after MLC are given. In addition, the stimulatory effects of LAA stimulation (**B**) vs. without LAA stimulation (**A**) on the provision of degranulating (leukemia-specific) immunoreactive cells and subsets in the WB of AML patients after MLC are compared. Given are the mean frequencies ± standard deviation (SD) of T/B cells and innate immune cells. Statistical significance was tested by a multiple-paired *t*-test. (n) number of cases; significance is defined as “highly significant” in cases with *p*-values ≤ 0.005, “significant” with *p*-values ≤ 0.05, “borderline significant” with p-values between 0.05 and 0.1, and “not significant” (n.s.) with *p*-values ≥ 0.1. Abbreviations of cell subpopulations are given in Table 1.

**Figure 6 ijms-25-06983-f006:**
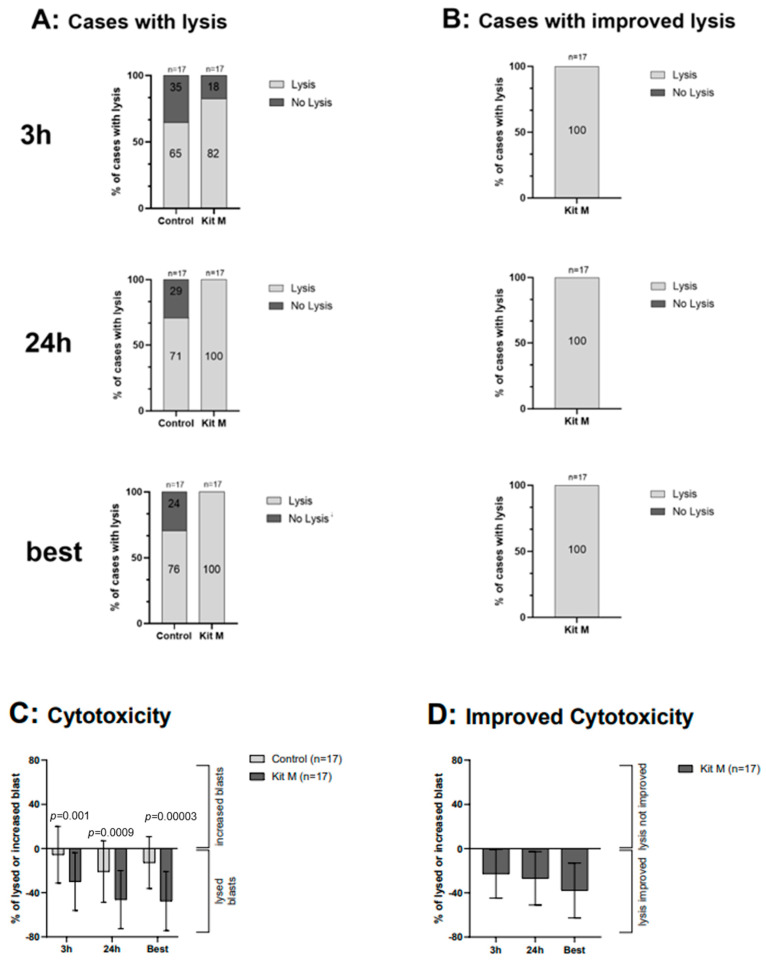
Effects of Kit M-treated WB on the antileukemic activity after MLC, detected via CTX. After MLC of Kit M-treated vs. untreated WB, these (‘immune effector’) cells were mixed with blast-containing MNCs (‘target cells’) and cultured for 3 h and 24 h. Given are the results after 3 h, 24 h, and the best achieved lysis after either 3 h or 24 h of the incubation time of the effector with target cells. The percentage of cases with lysis (**A**) and with improved lysis (**B**) compared to the control group (with target and effector cells mixed shortly before the measurement (Control)) is given. Given are the average ± standard deviation of achieved cytotoxicity (**C**) and improved cytotoxicity (**D**). Statistical significance was tested by a multiple-paired *t*-test. (n) number of cases; significance is defined as “highly significant” in cases with *p*-values ≤ 0.005, “significant” with *p*-values ≤ 0.05, “borderline significant” with *p*-values between 0.05 and 0.1. Abbreviations of cell subpopulations are given in Table 1.

**Figure 7 ijms-25-06983-f007:**
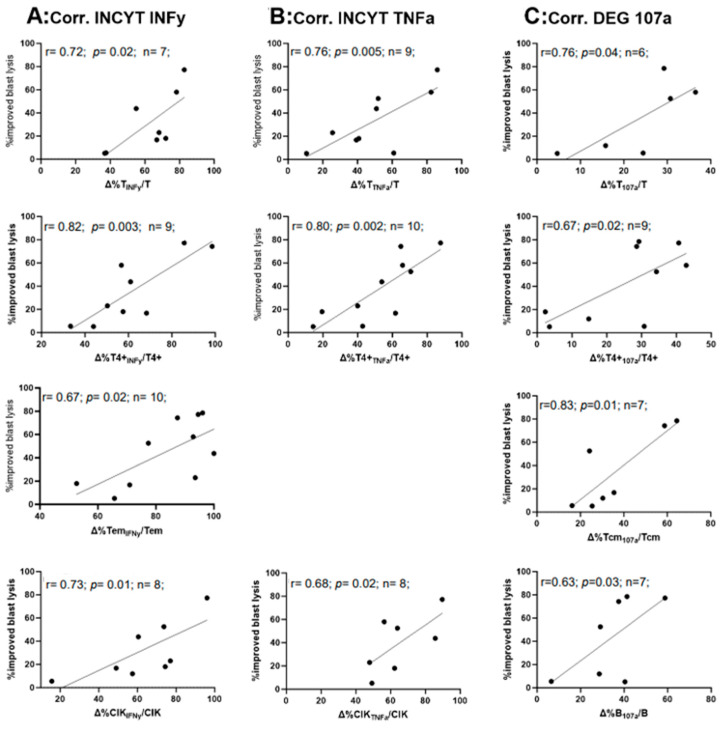
Correlation of the relative increase of IFNy- and TNFa-producing or degranulating cells with the relative improvement of blast lysis (=improved blast lysis) in Kit M-treated vs. untreated WB after MLC. Correlations of the IFNy (**A**) and TNFa (**B**) relatively to control produced immunoreactive cells (T, T4+, CIK; Tem, detected by INCYT) and of relatively to control produced CD107a positive cells ((**C**), detected by DEG), with the improved lysis compared to control, detected by CTX, are given. All frequencies of immunoreactive cell values are given as percentual differences (‘deltas’, (Δ%) of Kit M-treated vs. untreated control cultures. Lysis improvement refers to a relative increase in lysed blasts vs. control. Given are the (n) number of cases, evaluated by Pearson correlation analyses. Correlation coefficients (r) and *p*-values (one-tailed) are given. Significance is defined as “highly significant” in cases with *p*-values ≤ 0.005, “significant” with *p*-values ≤ 0.05, “borderline significant” with *p*-values between 0.05 and 0.1. Abbreviations of cell subpopulations are given in Table 1.

**Figure 8 ijms-25-06983-f008:**
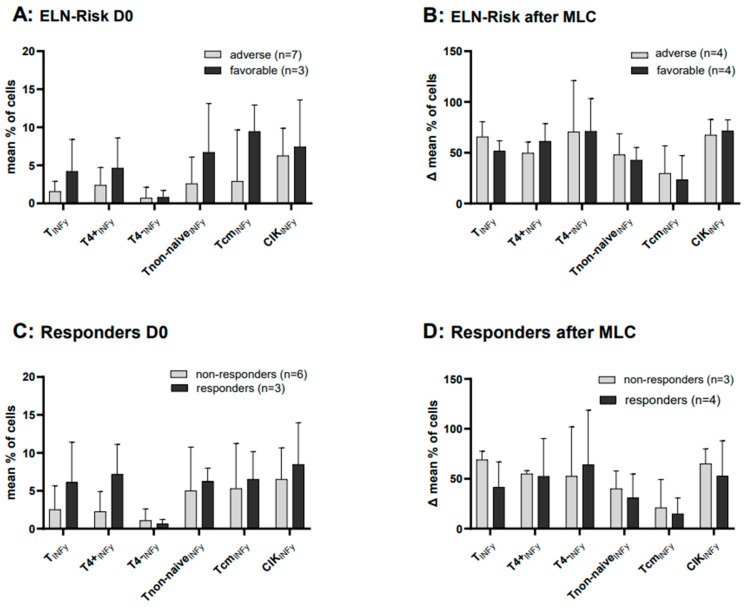
Frequencies of IFNy-producing cells in uncultured samples (left side) or in Kit M-treated (vs. untreated) WB after MLC (right side) with patients’ allocation to cytogenetic ELN risk groups and patients’ response to induction chemotherapy Frequencies of uncultured immunoreactive cell values in patients’ samples subdivided into ELN risk groups (**A**,**B**) or responders/non-responders to induction chemotherapy (**C**,**D**) are given. Frequences of immunoreactive cell values before and after MLC are given as percentual differences (‘deltas’, Δ%, of Kit M-treated (vs. untreated control) cultures of patients’ samples subdivided into ELN risk groups or responders/non-responders to induction chemotherapy. Given are the (n) number of cases. Abbreviations of cell subpopulations are given in Table 1.

**Table 1 ijms-25-06983-t001:** Cells and cell subsets as evaluated by flow cytometry.

Cell Type	Name of Subgroups	Abbreviation of Subgroups	Surface Marker	Referred to	Abbreviation	Reference
**Subtypes of blasts and DC**						
Blast cells	Leukemic blasts	Bla	Bla^+^ (e.g., CD34^+^, CD177^+^)	WB	Bla/WB	[26]
	Proliferating blasts	Blaprol-CD71	Bla^+^DC^−^CD71^+^	Bla	Blaprol-CD71/Bla	[26]
	Proliferating blasts	Blaprol-IPO38	Bla^+^DC^−^IPO38^+^	Bla	Blaprol-IPO38/Bla	[26]
Dendritic cells	Dendritic cells	DC	DC^+^ (CD80^+^, CD206^+^)	WB	DC/cells	[26]
	Leukemia derived DC	DCleu	DC^+^Bla^+^	WB	DCleu/cells	[26]
				Bla	DCleu/Bla	[26]
	Mature DC	DCmig	DC^+^CD197^+^	WB	DCmig/cells	[26]
	Mature DCleu	DCleu-mig	DC^+^CD197^+^Bla^+^	WB	DCleu-mig/cells	[26]
				DC	DCleu-mig/DC	[26]
				DCleu	DCleu-mig/DCleu	[26]
				Bla	DCleu-mig/Bla	[26]
**Subtypes of immune-reactive cells**						
T cells	CD3^+^ pan T cells	T	CD3^+^	WB	T/cells	[10]
	CD4^+^ coexpressing T cells	T4+	CD3^+^CD4^+^	T	T4+/T	[10]
	CD8^+^ coexpressing T cells	T4-	CD3^+^CD8^+^	T	T4-/T	[10]
	Proliferating T cells-early	Tprol-early	CD3^+^CD69+	T	Tprol-early/T	[10]
	Proliferating T cells-late	Tprol-late	CD3^+^CD71^+^	T	Tprol-late/T	[10]
	Non-naive T cells	Tnon-naive	CD3^+^CD45RO^+^	T	Tnon-naive/T	[8]
	Central (memory) T cells	Tcm	CD3^+^CD45RO^+^CD197^+^	T	Tcm/T	[8]
**Subtypes of different degranulating (CD107a^+^) cells**						
T cells	CD3^+^ pan T cells	T_107a_	CD107a^+^CD3^+^	T	T_107a_/T	[27]
	CD4^+^ coexpressing T cells	T4+_107a_	CD107a+CD3^+^CD4^+^	T	T4+_107a_/T	[10]
	CD8^+^ coexpressing T cells	T4-_107a_	CD107a+CD3^+^CD8^+^	T	T4-_107a_/T	[10]
	Non-naive T cells	Tnon-naive_107a_	CD107a^+^CD3^+^CD45RO+	Tnon-naive	Tnon-naive_107a_/Tnon-naive	[27]
	Effector (memory) T cells	Tem_107a_	CD107a^+^CD3^+^CD45RO^+^CD197^−^	Tem	Tem_107a_/Tem	[27]
	Central (memory) T cells	Tcm_107a_	CD107a^+^CD3^+^CD45RO^+^CD197^+^	Tcm	Tcm_107a_/Tcm	[27]
NK cells	CD3^−^CD56^+^ NK cells	NK_107a_	CD107a^+^CD3^−^CD56^+^	NK	NK_107a_/NK	[27]
CIK cells	CD3^+^CD56^+^ CIK cells	CIK_107a_	CD107a^+^CD3^+^CD56^+^	CIK	CIK_107a_/CIK	[27]
B cells	CD19^+^	B_107a_	CD107a^+^CD19^+^	B	B_107a_/B	[27]
**Subtypes of different intracellularly IFNy** or **TNFa-producing cells**						
T cells	CD3^+^ pan T cells	T_IFNy/TNFa_	IFNy^+^/TNFa^+^CD3^+^	T	T_IFNy/TNFa_/T	[8]
	CD4^+^-coexpressing T cells	T4+_IFNy/TNFa_	IFNy^+^/TNFa^+^CD3^+^ CD4^+^	T4+	T4^+^_IFNy/TNFa_/T4^+^	[8]
	CD8^+^-coexpressing T cells	T4-_IFNy/TNFa_	IFNy^+^/TNFa^+^CD3^+^CD8^+^	T4-	T4-_IFNy/TNFa_/T4-	[8]
	Non-naive T cells	Tnon-naive _IFNy/TNFa_	IFNy^+^/TNFa^+^CD3^+^CD45RO+	Tnon-naive	Tnon-naive_IFNy/TNFa_/Tnon-naive	[8]
	Effector (memory) T cells	Tem _IFNy/TNFa_/Tem	IFNy^+^/TNFa^+^CD3^+^CD45RO^+^CD197^−^	Tem	Tem _IFNy/TNFa_/Tem	[8]
	Central (memory) T cells	Tcm _IFNy/TNFa_	IFNy^+^/TNFa^+^CD3^+^CD45RO^+^CD197^+^	Tcm	Tcm _IFNy/TNFa_/Tcm	[8]
NK cells	CD3^−^CD56^+^ NK cells	NK_IFNy/TNFa_	IFNy^+^/TNFa^+^CD3^−^CD56^+^	NK	NK_IFNy/TNFa_/NK	[8]
CIK cells	CD3^+^CD56^+^ CIK cells	CIK_IFNy/TNFa_	IFNy^+^/TNFa^+^CD3^+^CD56^+^	CIK	CIK_IFNy/TNFa_/CIK	[8]

**Table 2 ijms-25-06983-t002:** Patients’ Characteristics.

Patient No.	Sex	Age at dgn	FAB type	Stage	Blast in PB (%) *	Blast Phenotype (CD)	ELN-Risk- Stratification	Response to Induction Chemotherapy	Conducted Experiments
**AML**									
P1509	m	60	pAML/M2	dgn	48	13,33,34,65,117	favorable	response	DCC, MLC, INCYT, CSA, CTX
P1514	m	68	SAML	dgn	51	33,56,117	favorable	no response	DCC, MLC, INCYT, DEG
P1518	f	83	pAML/M5	dgn	72	14,15,34,65	favorable	no response	DCC, MLC, INCYT, CSA
P1526	f	74	pAML	dgn	61	15,33,34,56,65,117	favorable	n.d.	DCC, MLC, DEG, CSA
P1602	m	67	pAML	dgn	16	13,33,34,56,117	favorable	response	DCC, MLC, CTX
P1630	m	29	pAML	dgn	16	13,15,33,34,64,117	Tavorable	response	DCC, MLC, INCYT, DEG, CTX
P1512	f	80	pAML	dgn	40	13,34,117	adverse	no response	DCC, MLC, INCYT, DEG
P1567	f	98	SAML	dgn	16	15,34,65,117	adverse	no response	DCC, MLC, INCYT, DEG, CTX
P1572	f	63	SAML	dgn	12	13,33,34,65,117	adverse	response	DCC, MLC, INCYT, DEG, CTX
P1573	m	61	pAML/M6	dgn	13	13,34,65,71,117	adverse	no response	DCC, MLC, INCYT, DEG, CTX
P1581	m	56	pAML/M4	dgn	58	13,15,33,34,117	adverse	no response	DCC, MLC, INCYT, DEG, CTX
P1638	m	68	sAML	dgn	33	4,13,33,34,56,117	adverse	no response	DCC, MLC, INCYT, DEG, CTX
P1527	m	42	pAML/M2	dgn	28	7,13,15,33,34,65,117	intermediate	n.d.	DCC, MLC, INCYT, DEG, CSA, CTX
P1604	f	60	sAML/MDS	dgn	16	7,13,33,34,65,117	intermediate	n.d.	DCC, MLC, INCYT, DEG, CTX
P1635	m	51	pAML	dgn	25	15,33,34,117	intermediate	response	DCC, MLC, INCYT, DEG
P1594	f	70	pAML/M4	Per. disease	20	13,33,34,65,117			DCC, MLC, INCYT, DEG, CTX
P1595	f	50	pAML	Per. disease	15	13,33,34,56,65,117			DCC, MLC, CTX
P1597	f	83	sAML	Per. disease	54	15,33,34,56,65,117			DCC, MLC, INCYT, DEG, CTX
P1603	f	32	pAML	Per. disease	50	15,33,34,117			DCC, MLC, INCYT, DEG, CTX
P1616	f	69	pAML	Per. disease	16	33,34,117			DCC, MLR, INCYT, DEG
P1497(2)	m	74	SAML	relapse	59	13,34,65,117			DCC, MLR, INCYT, DEG, CTX
P1516	f	52	SAML	relapse	70	15,33,34,65,117			DCC, MLR, INCYT, DEG, CTX
P1522	m	47	SAML	relapse	55	13,34,71,117			DCC, MLR, INCYT, DEG
P1598	f	61	sAML	relapse	25	13,33,34,117			DCC, MLR, INCYT, DEG, CTX
P1599	f	71	pAML/M4	relapse	79	7,13,33,34,117			DCC, MLC, DEG, CTX
P1632	f	56	pAML	relapse	65	13,33,34,56,65,117			DCC, MLC, INCYT, DEG, CTX
**HEALTHY**									
P1510	m	22							DCC, MLC, INCYT, DEG
P1513	m	27							DCC, MLC, INCYT, DEG
P1517	m	39							DCC, MLC, INCYT, DEG
P1523	m	17							DCC, MLC, INCYT, DEG, CSA
P1566	f	54							DCC, MLC, INCYT, DEG
P1579	m	30							DCC, MLC, INCYT, DEG
P1580	f	24							DCC, MLC, INCYT, DEG
P1582	m	27							DCC, MLC, INCYT, DEG
P1583	m	28							DCC, MLC, INCYT, DEG
P1585	m	29							DCC, MLC, INCYT, DEG
P1586	m	29							DCC, MLC
P1590	f	23							DCC, MLC
P1592	f	58							DCC, MLC, INCYT, DEG
P1596	m	26							DCC, MLC
P1611	f	27							DCC, MLC, INCYT, DEG
P1613	f	24							DCC, MLC, INCYT, DEG
P1636	m	22							DCC, MLC, INCYT, DEG
P1637	f	22							DCC, MLC, INCYT, DEG

Legend: f: female; m: male; AML: acute myeloid leukaemia; FAB classification: French American British classification of AML; WHO classification: world health organisation classification of AML; pAML: primary AML, sAML: secondary AML; MDS: MyeloDysplastic Syndrome; ELN: European Leukaemia Network; CD: Cluster of differentiation; bold: antibody used for expression analyses; WB: whole blood; dgn: diagnosis; Per. disease: persisting disease; DCC: dendritic cell culture measurements; MLC: mixed lymphocyte culture measurement; DEG: Degranulation Assay; INCYT: Intracellular Cytokine Assay; CSA: Cytokine Secretion Assay; CTX: cytotoxicity measurements; n.d.: no data; * indicates a clinical parameter on the day of sampling.

## Data Availability

Data are contained within the article.

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
