# Peer review of "Effective and Successful Quantification of Leukemia-Specific Immune Cells in AML Patients’ Blood or Culture, Focusing on Intracellular Cytokine and Degranulation Assays"

_ijms, 2024, doi:10.3390/ijms25136983_

Round 1

Reviewer 1 Report

Comments and Suggestions for Authors

Reviewer 2 Report

Comments and Suggestions for Authors

This is a very interesting and extensive work on quantification of leukemia specific immune cells in AML patients’ blood or culture settings.

Here are my remarks:

The title is very long. Perhaps the authors could try to make it more concise.

In the introduction there are some mistakes especially in the paragraph about the innate and adaptive immune system. The innate immune system is not consisted by the cells, but these cells belong to the innate immune system with a lot of other organs etc. The cells are not defined by only the markers mentioned here e.g. macrophages are not just CD15+, these markers are only examples. NK cells are CD56+CD3- and in fact they are CD56+/CD16+CD3- (and I don’t know whether the results obtained in the study using only CD56 and not CD16 are correct). At the end of the paragraph reoccurring antigen is not correct (the cells do not fight antigens). In general, a rephrasing of the introduction, paying a lot more attention is needed.

A part of the discussion concerning other methods that could be used in comparison to the ones used here should be part of the introduction and not of the discussion (p. 15).

The aims could be more general not including so many details of the methods used.

There is a big difference of age between patients and healthy controls. Perhaps a sentence is needed here. In table 1 blasts (as blasts is a morphological characterization, perhaps immature cells is a better definition for flow cytometry) phenotype (CD) should be positive CD)

In general, we write flow cytometry with two words. In 2.4., 2.8. some figures with the gating strategy could be shown (at least as supplementary information). In 2.6. the autologous T cells were isolated? Describe the method.

Regarding the results there is a lot of information and the figures are informative enough. Could the authors include a summary table of all the results or two, to facilitate the reader to understand the most important findings?

There are some English and typing errors.

Comments on the Quality of English Language

A careful inspection of the language and typing errors is needed.

Reviewer 3 Report

Comments and Suggestions for Authors

The article entitled “Intracellular Cytokine Assays in Combination With Degranulation Assay Contribute Significantly To Detect and Quantify Leukaemia Specific Immune Cells In AML Patients’Blood Or Culture Settings” is an original research addresses to investigate novel treatments in a hematologic disease characterized by severe prognosis such as Acute Myeloid Leukemia.  The authors studied 26 AML patients at diagnosis and after allogenic stem cell transplantation using a array of experimental laboratory tests such as the functional  assays INCYT, DEG and CTX and culture cells with Kit-M.  In summary, the results show that it is possible modulate the immune system in order to kill the leukemic cells and hence to provide a better prognosis.  I think that the project research is well-conducted in terms of methods and aim.

The authors focuses the attention on an important medical need regarding the treatment of acute myeloid leukemia (AML), the most common acute leukemia in adult, characterized by up to 50% of relapse after initial chemotherapy and prognosis poor.   There are several targeted immunotherapies, such as monoclonal antibodies, bispecific T (TCE) and killer cell engager molecules, and chimeric antigen receptor (CAR) T cells, under clinical evaluation.   In this article, the authors addresses their research about the possible anti-leukemia immune response of the dendritic cells (DCs).   Therefore, they generated DCs and leukamia-derived DCs by WB treated with GM-CSF and analyzed the DCs by flowcytometry and used specific assays such as degranulation assay (DEG) and intracellular citokine assay (INCYT) as well as cytotoxicity fluorolysis assay (CTX).   The used tests and results are appropriately described in tables.   DCs-based therapy may provide new treatment perspectives and a safer alternative for targeting AML cells, However, a complication frequently associated with immune therapies is the cytokine release syndrome or neurotoxicity.   In addition, potent cytotoxicity may be associated with toxicity in patients.   I think that the authors should day something about this concepts in the section “Discussion”.   Anyway, I think that this article is suitable for publication with the addition of these comments.

Round 2

Reviewer 1 Report

Comments and Suggestions for Authors

The authors managed to modify the MS, which I will accept in the present form.